

# Automated programming approaches to enhance computer-aided translation accuracy

Tao Zhao[1,2] and Mazni Binti Alias[1]

[1] Faculty of Management, Multimedia University, Cyberjaya, Malaysia
[2] School of International Education, Henan University of Engineering, Zhengzhou, Henan, China

## ABSTRACT

With the continued development of information technology and increased global cultural exchanges, translation has gained significant attention. Traditional manual translation relies heavily on dictionaries or personal experience, translating word by word. While this method ensures high translation quality, it is often too slow to meet the demands of today's fast-paced environment. Computer-assisted translation (CAT) addresses the issue of slow translation speed; however, the quality of CAT translations still requires rigorous evaluation. This study aims to answer the following questions: How do CAT systems that use automated programming fare compared to more conventional methods of human translation when translating English vocabulary? (2) How can CAT systems be improved to handle difficult English words, specialised terminology, and semantic subtleties? The working premise is that CAT systems that use automated programming techniques will outperform traditional methods in terms of translation accuracy. English vocabulary plays a crucial role in translation, as words can have different meanings depending on the context. CAT systems improve their translation accuracy by utilising specific automated programs and building a translation *corpus* through translation memory technology. This study compares the accuracy of English vocabulary translations produced by CAT based on automatic programming with those produced by traditional manual translation. Experimental results demonstrate that CAT based on automatic programming is 8% more accurate than traditional manual translation when dealing with complex English vocabulary sentences, professional jargon, English acronyms, and semantic nuances. Consequently, compared to conventional human translation, CAT can enhance the accuracy of English vocabulary translation, making it a valuable tool in the translation industry.

## INTRODUCTION

The frequent exchanges of world culture, trade, and economy have become one of today's most popular industries. As the current universal language in the world, English translation must be taken seriously. In the 1980s, English translation was manual, and translators relied on dictionaries to translate word by word. However, the translation accuracy was high, greatly affecting the speed. Computer translation has become a popular

Corresponding author
Tao Zhao, zhaoxy0628@163.com

method with the development of information technology. Computer translation is divided into machine translation and computer-assisted translation. Machine translation greatly improves translation speed through direct translation of vocabulary. The amount of translation in 1 s is the translator's workload in a day, but the quality of translation could be more satisfactory. Computer-assisted translation can effectively improve translation quality through unique translation memory technology. Set up an automatic program for computer-aided translation and let the computer-aided translation translate according to a specific program. For example, when it is detected that the translated word appears above, the meaning of the word can be obtained so that the contextual translation is consistent. Special English translation scenarios require accurate English translation. Testing the accuracy of computer-assisted translation based on automatic programming can help understand the quality comparison between machine-assisted translation and human translation so that machine-assisted translation can be applied to suitable fields and improve translation efficiency. Therefore, it has research significance.

Human translation is mainly based on translation quality, but the translation takes too long. Computer-assisted translation with machine-assisted translators can improve translation efficiency. Among them, *Donovan & Ledgard*'s *(2017)* research showed that computer-assisted translation can effectively shorten the translation time. *Xu & Li (2021)* translated English through computer-assisted translation, and the accuracy of the translation of English vocabulary was very high. *Cattoni et al. (2021)* built an English term *corpus* for computer-aided translation, which improved the accuracy of English term translation. *Wu*'s *(2018)* research pointed out that computer-aided translation was an effective English translation and had a wide range of applications in dealing with professional English. *Guo (2018)* tested the translation of English acronyms by computer-assisted translation through experiments, and computer-assisted translation had a translation accuracy of nearly 100%.

Although computer-assisted translation can improve the efficiency of English translation, more research should be done on automatic programming. Automatic programming is the research direction of intelligent informatization. Applying automatic programming to computer-aided translation can effectively improve translation quality. *Strmecki, Magdalenic & Radosevic*'s *(2018)* research showed that automated programming could help computer-aided translation effectively translate contextually, consistently improving translation accuracy. *Li & Zhong*'s *(2020)* research pointed out that the automatic program can translate different words more targetedly by controlling the computer-assisted translation. *Velasquez (2018)* designed an automatic program for computer-aided translation of English professional vocabulary and improved translation accuracy through *corpus* analysis. *Makihara et al. (2018)* pointed out that designing a complete and comprehensive automatic program can improve the accuracy of computer-assisted translation in translation. *Ahmadi & Abadi*'s *(2020)* research pointed out that object-oriented feature design can improve the performance of automatic programming. Although automated programmed computer-assisted translation can improve translation accuracy, comparative analysis with human translation accuracy still needs to be improved. Computer-aided translation establishes a translation *corpus* through translation

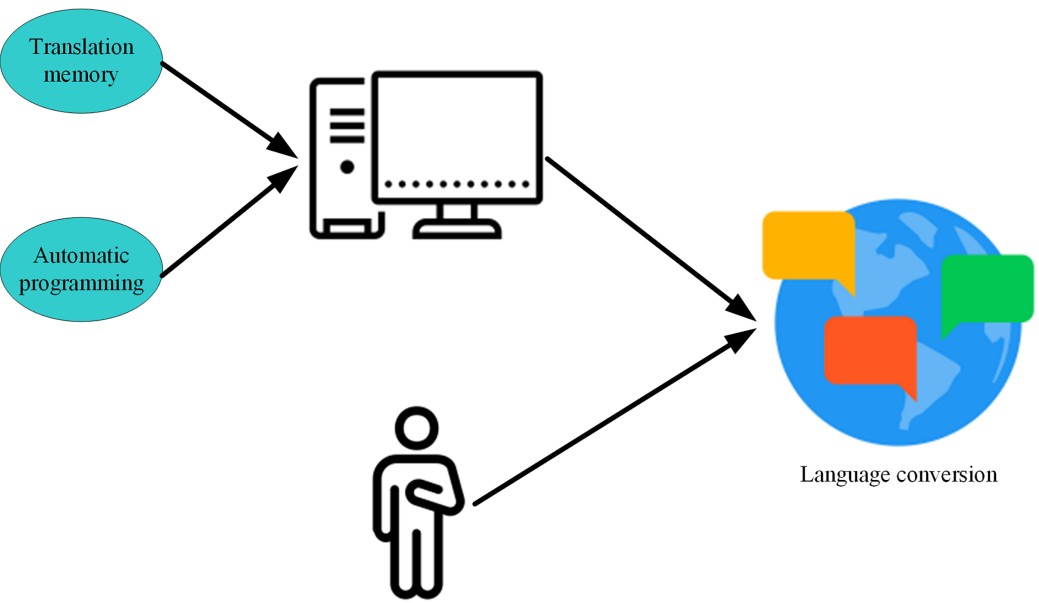

**Figure 1 The computer-aided translation model.**

memory technology, which improves the understanding of polysemy in English vocabulary during translation and can be well combined with the context of translation for translation. Automated programming can effectively improve the analysis of *corpus* by computer-aided translation and improve the accuracy of translation. The innovation of this method is to compare and analyse English vocabulary using computer-assisted translation based on automatic programming and traditional manual translation.

## METHODS OF COMPUTER-ASSISTED TRANSLATION

International cultural exchanges are becoming increasingly frequent, and more attention is paid to converting between languages (*Screen, 2017*). Traditional manual translation is unsuitable for today's translation market due to its slow speed. Computer-aided translation, a model of machine-assisted translation, does not fully replace human input but helps translators increase their speed through programmatic support. The model of computer-assisted translation is shown in Fig. 1.

In Fig. 1, computer-assisted translation is a mode in which humans and computers cooperate to translate. The computer programs the text to be translated automatically. Using translation memory, it provides the translator with accurate translation results of 1 qds, which can improve translation efficiency. In this study, installing the CAT system required the utilization of various essential software tools and technologies. The principal CAT program utilised was Trados Studio, renowned for its strong translation memory management capabilities and extensive file format support. Trados Studio was selected based on its sophisticated capabilities, which enable effortless integration with translation memories and machine translation engines. In addition, Moses served as a statistical machine translation (SMT) engine, offering a flexible platform for conducting experiments with different translation models. Moses enabled us to optimise the translation process by

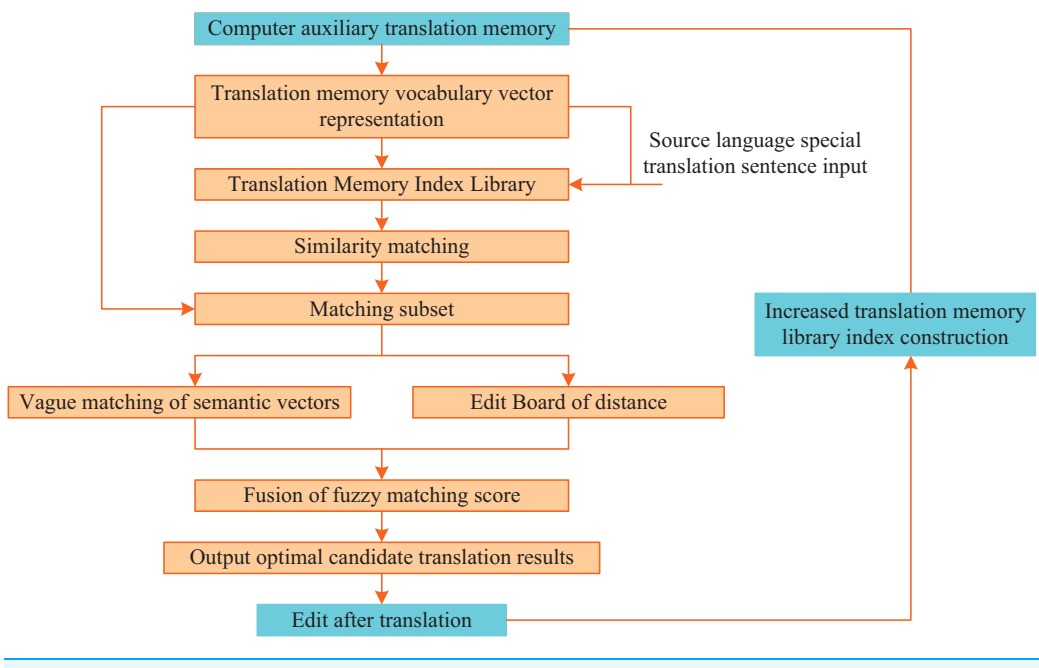

**Figure 2 Schematic figure of translation memory.**

modifying parameters and training the system on particular corpora that are pertinent to our research.

## Translation memory technology

Translation memory is the core of computer-assisted translation and is a data storage mode. It saves many previously translated words and rules during translation, effectively reducing the time needed to translate the same sentence, making translation consistent, and improving translation efficiency (*Bhattacharjee, 2018*). The principle of translation memory is shown in Fig. 2.

In Fig. 2, translation memory builds English vocabulary vectors, and each English vocabulary vector contains multiple corresponding translations of the English vocabulary and translations in different foregrounds. However, when English vocabulary is encountered during English translation, it will search for the matching vocabulary translation in the translation memory through information retrieval and filter the complete translation through matching. The content translated will also be saved in the translation memory. In the translation process, it is often encountered with a large amount of repeated text or paragraphs. If people translate an article and then re-translate it, it will waste workforce and material resources. The translation memory can save a large amount of translation content. When the matching degree between the current translation content and the translation memory is high, the translation at that time will be available for the translator to choose, which can effectively avoid repeated translations (*Negri et al., 2017*). Translation memory can also perform professional sentence management. In English translation, many professional English words represent different meanings, which cannot be translated according to the mode of spoken language. The translation must be accurate,

and the same English vocabulary in the translation context must be consistent. Translation memory can identify professional sentences and give relevant explanations, and the meaning of professional sentences can be directly obtained by retrieving translation memory during translation (*Sanchez-Gijon, Moorkens & Way, 2019*). The study utilises the CAT system incorporating neural machine translation (NMT) algorithms. These algorithms are highly proficient in managing intricate linguistic patterns using deep learning methodologies. The translation memory is administered through phrase-based models and statistical alignment methods, enhancing the system's capacity to retrieve previously translated segments and maintain terminological consistency. These technologies enable the CAT system to adjust to the precise style and demands of professional translations, enhancing its accuracy in managing specialised content.

The incorporation of translation memories into the CAT system took place through a series of stages: The incoming text was automatically segmented into sections that could be compared to the stored translations in the TM database. The system employed methods, such as fuzzy matching, to detect segments that had a significant level of resemblance to those stored in the translation memory. After identifying a suitable match, the system presented the translator with a pre-existing translation, which the translator might subsequently approve, alter, or decline depending on the surrounding circumstances. The translation memory was constantly updated with new translations, thereby improving the system's capacity to offer precise suggestions as time went on. The utilisation of translation memory technology had a substantial effect on the correctness of translations. Using a large collection of previous translations decreased the chances of using inconsistent language and lowered errors while translating intricate or specialised content.

## Automatic programming

With the continuous updating of information technology, intelligent and automated programs are applied to various fields such as medical care, education, and scientific computing. Automated programming greatly improves the translation process by simplifying and optimising multiple processes that would otherwise require a lot of manual effort. In contrast to manual programming, automatic programming relies on established structures and algorithms to perform tasks with minimal human interaction, eliminating the need for humans to code, debug, and refine each translation rule or algorithm. This automated technique enables the system to efficiently analyse and handle enormous amounts of text by breaking down intricate phrases into more manageable parts, which can then be compared with pre-existing translation memories. Automatic program design has the advantages of automatic execution, high execution accuracy, and fast execution speed, and it can be well applied to computer-assisted translation. Automatic programming is a combination of the program's structure and the algorithm. The program structure is the foundation, the algorithm is the design idea, and the program structure is the splitting of the problem structure. Let a program:

$$y = (3 * (\sin 3x) + 4) / \cos 2x. \tag{1}$$

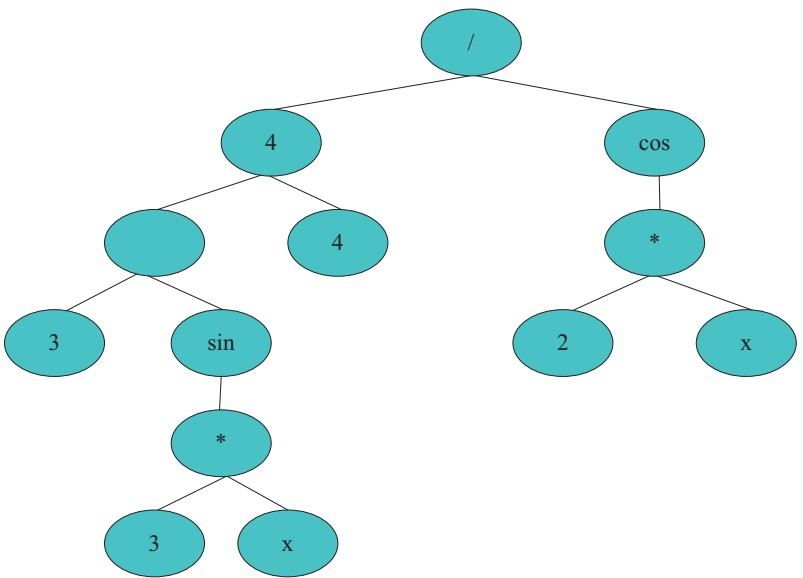

**Figure 3  Program structure.**     

Then, the program structure corresponding to Formula (1) is shown in Fig. 3.

In Fig. 3, the execution process of the automatic program structure must be executed according to the set rule, which is a sequential execution process. When dealing with complex English vocabulary in computer-aided translation, it is essential to assess the accuracy of the translation. In automatic programming, long and complex sentences are broken down into simpler terms that can be matched with entries in the translation memory, and the English words are then processed using fuzzy matching techniques (*Chang et al., 2017*). Let the constraint parameter of the divided English vocabulary be, where M is the correct matching degree of the divided vocabulary, and U is the accuracy evaluation set of complex vocabulary translation.

$$\Delta(R) = \begin{cases} b_k, k = g(R) \\ c_k = R - k, c_k \in [-0.5, 0.5). \end{cases} \tag{2}$$

In Formula (2), g represents the English vocabulary translation matching operator. To predict the accuracy of computer-assisted translation of complex English sentences, the mapping of complex English sentences is expressed as:

$$\Delta : [0, M] \rightarrow U \times [-0.5, 0.5). \tag{3}$$

Multi-semantic analysis of complex sentences:

$$(\bar{b}, \bar{c}) = \Delta \left( \sum_{i=1}^{n} \frac{1}{n} \Delta^{-1}(b_i, c_i) \right). \tag{4}$$

Search all complex sentences in the translation and output the translated results:

$$(\overline{b}, \overline{c}) = \Delta\left(\sum_{i=1}^{n} h_i \Delta^{-1}(b_i, c_i)\right). \tag{5}$$

In Formula (5), $\overline{b} \in U\overline{c} \in [-0.5, 0.5] \sum_{i=1}^{n} h_i = 1$. For the case of multi-semantics in English vocabulary translation, let the multi-semantics be expressed as P, Q, then the similarity of complex sentences before translation can be expressed as:

$$F(P, Q) = \frac{C(P) \cdot C(Q)}{|C(P)| \cdot |C(Q)|}. \tag{6}$$

In Formula (6), F represents the similarity of English vocabulary before and after translation. Adaptive matching is performed according to the position of complex sentences in the translation file, and the most accurate translation sentences are used to replace them by analyzing the context (*Zivkovic, Tangelder & Kerkhoff, 2017*).

The *corpus* used to evaluate the CAT systems in this study was carefully selected to ensure a comprehensive assessment of translation accuracy. The *corpus* comprised around 500,000 words from various materials, such as technical manuals, legal documents, medical writings, and general literature. Diverse content was selected to accurately represent the various challenges that CAT systems may encounter in real-world scenarios, ranging from technical terminology to everyday language. The *corpus* selection criteria prioritized achieving a well-rounded representation of various text genres, with specific emphasis on including documents with intricate sentence patterns, idiomatic idioms, and specialised vocabulary.

The evaluation of translation correctness in this study included the use of three metrics: BLEU (Bilingual Evaluation Understudy) score, METEOR (Metric for Evaluation of Translation with Explicit ORdering), and TER (Translation Edit Rate). The selection of these measures was based on their extensive recognition in the field of natural language processing and their capacity to encompass all facets of translation quality. The metrics used in this study give a comprehensive evaluation framework that aligns with the study's aims. They examine both the correctness of the translations at a surface level and the semantic fidelity at a deeper level, which is the system's ability to convey the intended meaning accurately.

## COMPUTER-AIDED TRANSLATION EXPERIMENT DESIGN AND DATA

### Experimental data

Computer-aided translation based on automatic programming can significantly enhance translation speed and reduce the need for translators to repeatedly translate identical sentences by utilizing translation memory and automated translation programs (*Daemei & Safari, 2018*). However, computer-aided translation not only guarantees speed but also the quality of the translation. The accuracy of language translation is prioritized over translation speed. To analyse the accuracy of the computer-aided translation, a

**Table 1** The survey results affecting the accuracy of computer-aided translation.

| Serial number | Index | Number of people | Proportion |
| --- | --- | --- | --- |
| 1 | Complex sentence | 180 | 90% |
| 2 | Technical English vocabulary | 184 | 92% |
| 3 | Acronym | 178 | 89% |
| 4 | Semantics of English vocabulary | 168 | 84% |
| 5 | Simple English vocabulary | 146 | 73% |

**Table 2** Correlation analysis results.

| Serial number | Index | Relevance |
| --- | --- | --- |
| 1 | Complex sentence | 0.24 |
| 2 | Technical English vocabulary | 0.25 |
| 3 | Acronym | 0.22 |
| 4 | Semantics of English vocabulary | 0.23 |
| 5 | Simple English vocabulary | 0.06 |

questionnaire survey was conducted among 200 English vocabulary translation translators to investigate the factors that they thought could evaluate the accuracy of the computer-aided translation, the number of people and proportions in favour of each indicator. The results of the survey to evaluate the accuracy of computer-aided translation are shown in Table 1.

Table 1 lists five factors that may influence the accuracy of computer-aided translation. Among them, the highest approval rate for the technical English vocabulary index is 92%, and the average approval rate for the above five indicators is 85.6%. These five indicators are the factors that English translators think can affect the accuracy of English vocabulary translation (*Sun et al., 2020*; *Alsohybe, Dahan & Ba-Alwi, 2017*). A correlation analysis of English translation accuracy is conducted using the five indicators in Table 1. The purpose of the correlation analysis is to minimize the impact of extreme factors on translation accuracy. Experimental analysis is optional when the index is not related to the accuracy of English translation. The results of the correlation analysis for English translation accuracy based on the five indicators are presented in Table 2.

In Table 2, the most relevant index is the technical English vocabulary index, 0.25, followed by the complex sentence index, the correlation is 0.24, and the least relevant is the simple English vocabulary index, which is only 0.06. Therefore, the analysis of the simple English vocabulary index was excluded (*Su et al., 2017*).

## Experimental design

To explore the accuracy of computer-assisted translation based on automatic programming, it is necessary to compare the accuracy of translation with traditional manual translation. An experimental group was established for computer-assisted

**Table 3 Indicator validity analysis results.**

| Serial number | Index | Control group | Test group |
|---|---|---|---|
| 1 | Complex sentence | 84.6% | 86.8% |
| 2 | Technical English vocabulary | 86.6% | 92.2% |
| 3 | Acronym | 78.4% | 78.2% |
| 4 | Semantics of English vocabulary | 82.6% | 84.2% |
| 5 | Average | 83.1% | 85.4% |

translation, while a control group was assigned for human translation. The study involved a control group of conventional human translators, chosen for their expertise and familiarity with the CAT system. Both groups were provided with identical source texts and instructed to work without external aids. They were given a predetermined duration and required to adhere to industry standards for accuracy. The aim was to establish a standard for comparing human and automated translations under similar conditions. Compare the difference in translation accuracy of the two modes under the same translation lexical environment (*Donovan & Ledgard, 2017*). To determine if the first four indicators in Table 2 are effective for comparing translation accuracy between the two modes, a validity analysis of the relevant indicators is required.

The sample sizes used in our research were carefully chosen to ensure that the results are statistically significant and accurately represent the larger population. We have chosen a *corpus* consisting of around 500,000 words, sourced from various materials such as technical manuals, legal documents, and popular literature. This selection allows us to address a broad spectrum of linguistic difficulties. The extensive and diverse dataset offers a strong foundation for evaluating the performance of the CAT system in various settings. In addition, the experimental design consisted of 2,000 English vocabulary groups used as samples, with 1,500 groups allocated for the test set and 500 groups for the validation set. The researchers employed a four-fold cross-validation technique to ensure the reliability of the findings. The chosen sample sizes are sufficient to capture the variability in translation accuracy across different text types and translation contexts. The results of the validity analysis of the accuracy of English vocabulary translations under the two modes are shown in Table 3.

Table 3 shows that these four indicators are highly effective in assessing the accuracy of English vocabulary translation in both modes. Among them, these four indicators have an influence rate of 83.1% on the accuracy of English vocabulary in traditional manual translation and an influence rate of 85.4% on the accuracy of computer-aided translation of English vocabulary based on automatic program design. Therefore, the four indicators in Table 3 can be used to evaluate the accuracy of English vocabulary translation in the two modes (*Hong et al., 2019*; *Wong et al., 2020*). In the experimental group, the computer-aided translation system, designed through programming, retrieves the English vocabulary to be translated from the translation memory. The similarity before and after translation is

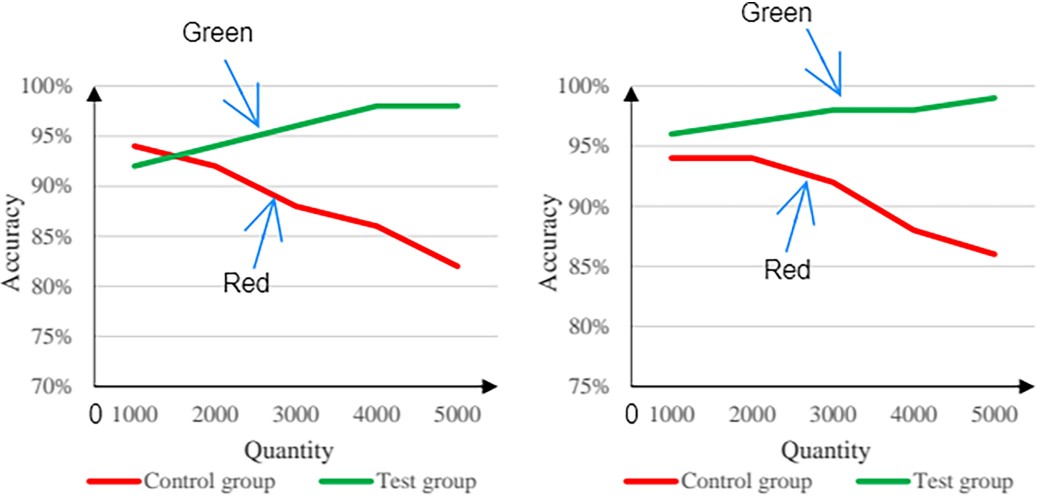

**Figure 4 The accuracy comparison result of English complex sentence translation.**

measured using Formula (6). When the similarity is high, the time of repeated translation can be saved, and the accuracy of the translation can be guaranteed.

# RESULTS ANALYSIS

Comparison of English complex sentence translation: Translating complex English sentences is often challenging for translators. The grammar in complex English sentences is nested layer by layer. To ensure the accuracy of English translation, the speed of manual translation of complex sentences is extremely slow (*Wang et al., 2016*). Computer-aided translation based on automatic programming can decompose complex sentences through translation memory technology and perform similarity analysis with translation memory. Since the length of complex sentences can impact the accuracy of English translation, the study analyzes translation accuracy by comparing long and short complex sentences. The experiment includes English complex sentences of varying lengths, categorized by every 1,000 English words. The accuracy of translations for each category is tested in both modes. We used t-tests to determine the statistical significance of the differences seen when comparing the accuracy of translations generated by the CAT system to traditional manual approaches. The selection of this test was based on its appropriateness for comparing the means of two groups. Furthermore, confidence intervals (95%) were computed to offer an approximation of the region where the actual disparity in translation accuracy exists. The statistical analyses provide more evidence that computer-assisted translation (CAT) systems, which employ automated programming, show a statistically significant enhancement in translation accuracy.

The comparison results of the accuracy of English complex sentence translation under the two modes are shown in Fig. 4.

Figure 4 shows that the English translation in both modes is more accurate for short complex sentences compared to long complex sentences. But in general, the accuracy rate of computer-assisted translation in translating complex sentences is higher than that of

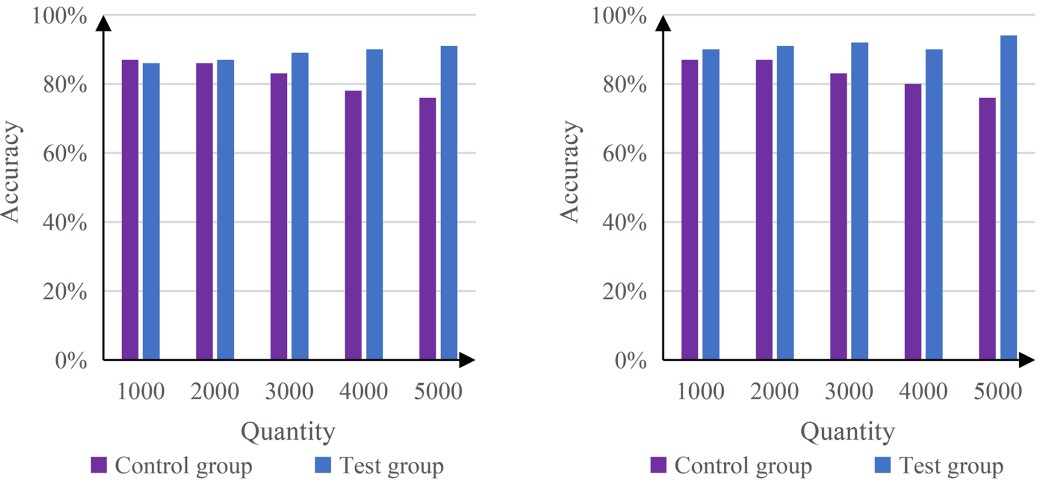

**Figure 5 The comparison result of the translation accuracy of scientific and technological English vocabulary.**

traditional human translation. Only when translating long, complex sentences and the number of English words is 1,000 is the accuracy of computer-assisted translation of complex English sentences lower than that of human translation. At this time, the accuracy rates of the two modes are 92% and 94%, respectively. The average accuracy rates for computer-assisted translation using automatic programming and traditional human translation of complex English sentences are 88% and 95%, respectively. Therefore, computer-aided translation based on automatic programming can improve the accuracy of complex English sentences (*Yan & Lei, 2020*). Comparison of vocabulary translation in English for science and technology: Professional English vocabulary is very professional, and technical English is the most common and difficult professional English vocabulary to translate. Human translation of technical English vocabulary often requires translators to understand the relevant majors deeply, and translation deviations will occur if the understanding is improper. Comparing the accuracy of technical English vocabulary translation between computer-assisted translation based on automatic programming and traditional manual translation. The accuracy of the two translation modes is observed by increasing the amount of technical English vocabulary translated. The technical English vocabulary of different lengths is studied separately. Figure 5 compares the accuracy rates of the two translation modes for technical English vocabulary translation.

In Fig. 5, when translating a long technical English sentence with a vocabulary of 1,000, the accuracy of computer-aided translation is 86%. The accuracy rate of traditional manual translation is 87%, which is similar. However, when the translated technical English vocabulary increases, traditional manual translation's accuracy decreases, while computer-assisted translation's accuracy increases. The translation accuracy of the two is 91% and 76%, respectively, when the vocabulary is 5,000. When translating short technical English, the accuracy rate of computer-assisted translation is higher than that of traditional human translation. The average accuracy of computer-assisted translation of technical English is 90%, and the average accuracy of traditional human translation of technical English is 82%.

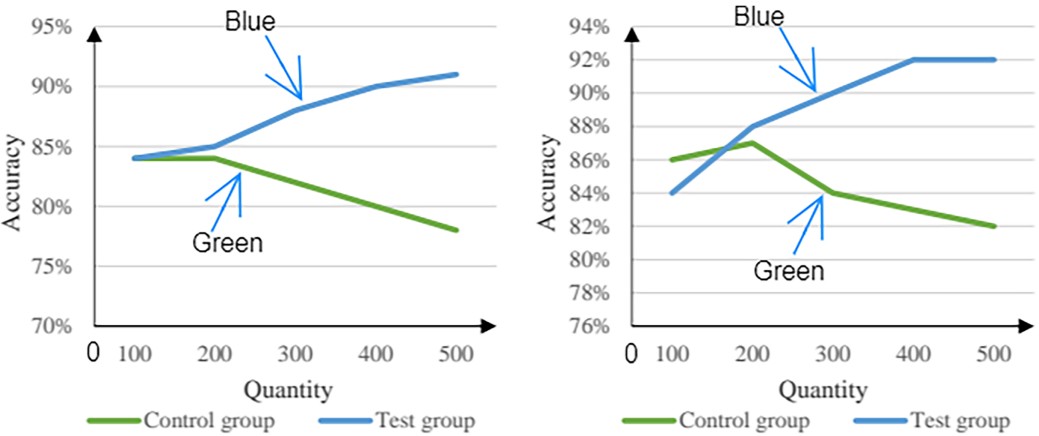

**Figure 6 Accuracy comparison result of English acronym translation.**

Therefore, computer-aided translation can improve the accuracy of technical English translations. Comparison of English acronym translations: English acronyms are also common in English translation. Because it's an acronym, it's hard to figure out what the acronym stands for, and the longer the acronym, the more difficult it is to translate. A separate discussion of acronyms of different lengths is required to compare the translation of English acronyms by the two translation modes (*Rosin et al., 2021*; *Zic & Zic, 2020*). Since acronyms generally occur less frequently, the increment for translating English acronyms is set to 100. Figure 6 presents the comparison results for the accuracy of the two translation modes in translating English acronyms.

In Fig. 6, when the number of translated English abbreviations increases, the accuracy of translating English acronyms by computer-aided translation based on automatic programming steadily improves. However, the accuracy of English acronym translation using traditional manual methods consistently declines. When the number of translated acronyms reaches 500, the accuracy rates of the two modes in translating long acronyms are 81.6% and 87.6%, respectively. The average accuracies of the two modes for translating short acronyms are 84.4% and 89.2%, respectively. The average acronym translation accuracy of the two translation modes is 86% and 90%. Therefore, computer-assisted translation based on automatic programming can improve the translation accuracy of English acronyms by an average of 4%.

Comparison of semantic translation of English vocabulary: In English vocabulary translation, the same English vocabulary often expresses different meanings in different contexts. The semantic translation of English vocabulary needs to be translated in conjunction with the understanding of the context, and it is easy to make mistakes in translation. Computer-aided translation based on automatic programming relies on translation memory to store context and provide possible semantics to translators for reference and translation. Comparing the semantic translation of English vocabulary between the two translation modes requires a gradient test on the translation's vocabulary size, and the translation's accuracy is tested for every 1,000 additional English words. The

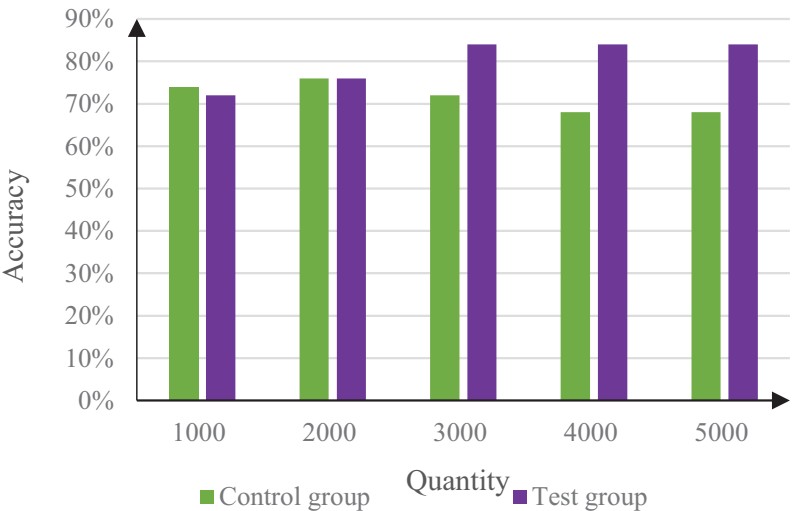

**Figure 7 Average wind speed and maximum wind speed changes under the influence of different atrium interface ratios.**

comparison results of the translation accuracy of English lexical semantics under the two translation modes are shown in Fig. 7.

In Fig. 7, the translation accuracy of the two translation modes is similar at the beginning, and the accuracy is 74% and 72%, respectively. However, when the translated English vocabulary increases, the translation memory of computer-assisted translation based on automatic programming continues to expand the data, and the translation accuracy of English vocabulary semantics continues to improve, with an average accuracy of 80%. In traditional manual translation, due to the increase in English vocabulary, the semantics of English vocabulary become more complex, and the translation accuracy of English vocabulary begins to decrease, with an average accuracy rate of 71.6%. Therefore, computer-aided translation based on automatic programming can effectively improve the translation of English vocabulary and semantics with the help of translation memory technology.

Although this study offers valuable insights into the efficacy of CAT systems that employ automated programming, several limitations must be recognized. Potential biases include the *corpus* selection, which, despite its diversity, may not fully represent all possible translation scenarios, particularly those involving highly specialised or rare language constructs. Additionally, the control group's skilled translators may have established a higher baseline for human translation accuracy, which could potentially have influenced the comparative results.

## Common errors in CAT and human translations

The study compares CAT systems and human translators, revealing recurring mistakes and limitations. CAT systems often generate word-for-word translations, while human translators make typographical errors, missing words, and conflicting terminology. Subjectivity in interpretation is also evident. The analysis suggests areas for improvement,

such as enhancing contextual understanding and consistency in terminology. The study also suggests combining the strengths of both systems and human translators for maximum translation accuracy.

Future advancements in CAT systems include integrating neural networks with deep learning capabilities, incorporating context-aware translation models, and exploring artificial intelligence techniques like reinforcement learning. These advancements could improve translation accuracy and precision in industries like legal and medical translations. Multilingual CAT systems could enhance consistency and reduce the need for multiple translation passes. Combining rule-based and statistical methods, hybrid models could improve accuracy in high-precision domains. Additionally, future research should explore the ethical implications of automated translation systems to ensure inclusivity and fairness in global communication.

The complexity of the original text, including specialised terminology and abbreviations, significantly impacts the precision of translation for both CAT systems and human translators. CAT systems struggle with specialised language due to a lack of familiarity with the terminology, leading to inaccurate interpretations in fields like medicine, law, or engineering. However, CAT systems can consistently employ accurate language, often surpassing human performance. Human translators excel in handling context-sensitive content, but CAT systems can surpass human performance in high-continuity domains.

## CONCLUSION AND FUTURE DIRECTIONS

Computer-aided translation based on automatic program design takes translation memory as the core, through designing automatic programs to search English vocabulary in translation memory to improve translation efficiency. Comparing computer-assisted and traditional human translations in English vocabulary by comparing the complex sentences of English vocabulary, technical English, acronyms, and the semantics of English vocabulary. The experimental results show that computer-aided translation based on automatic programming can improve the accuracy of complex English sentences by 7% and can effectively improve the accuracy of technical English, acronyms, and lexical-semantic translation. Computer-assisted translation based on automatic programming can improve translation speed and ensure quality. When the translated English vocabulary increases, the effect of computer-aided translation is better. Computer-assisted translation can improve human translation accuracy and help translators translate more efficiently. Future research should further enhance the contextual understanding and adaptability of CAT systems, particularly in translating idiomatic expressions and culturally nuanced content. Additionally, exploring the ethical implications of increasingly automated translation processes will be crucial as these technologies become more widespread. Overall, this study contributes to the ongoing development of CAT systems and provides a foundation for future advancements in translation technologies.

Future research directions could explore enhancing computer-assisted translation (CAT) systems by incorporating advanced machine learning algorithms, such as deep learning and neural networks, to better understand and handle semantic nuances and

specialized terminologies. Additionally, research could focus on improving real-time translation capabilities and evaluating CAT systems across multiple languages and complex sentence structures to further enhance their accuracy and reliability in diverse contexts.

### Funding
The authors received no funding for this work.

### Competing Interests
The authors declare that they have no competing interests.

### Author Contributions
- Tao Zhao conceived and designed the experiments, performed the experiments, analyzed the data, performed the computation work, prepared figures and/or tables, authored or reviewed drafts of the article, and approved the final draft.
- Mazni Binti Alias conceived and designed the experiments, performed the experiments, analyzed the data, performed the computation work, prepared figures and/or tables, authored or reviewed drafts of the article, and approved the final draft.

### Data Availability
   The code and dataset are available in the Supplemental Files.

### Supplemental Information
Supplemental information for this article can be found online at http://dx.doi.org/10.7717/peerj-cs.2396#supplemental-information.

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
