# Peer review of "Automated programming approaches to enhance computer-aided translation accuracy"

_PeerJ Computer Science, doi:10.7717/peerj-cs.2396_

## Round 0.1 · original submission · Major Revisions

Dear authors

Your manuscript has been reviewed with interest by the experts in the field and you can see that they are advising lot of improvement suggestions, therefore I endorse their input and also suggest the following to incorporate before you resubmit the article

1. Who are the targeted audience who can benefits from this research?

2. Clearly state the need for the research and the motivation behind it.

3. Improve the technical language of the manuscript.

4. Is the proposed approach applicable to one language for translation or is it a generic approach that is applicable to all languages?

Thank you

**Language Note:** The review process has identified that the English language must be improved. PeerJ can provide language editing services - please contact us at [email protected] for pricing (be sure to provide your manuscript number and title). Alternatively, you should make your own arrangements to improve the language quality and provide details in your response letter. – PeerJ Staff

Reviewer 1 ·

Basic reporting

1- The quality of the figures presented in the paper is substandard. The authors should enhance the clarity and resolution of all graphical elements to ensure they effectively convey the intended information.

2- The introduction section lacks a clear problem formulation. The authors need to explicitly define the problem they are addressing to provide context and focus for their research.

3- The literature review is insufficient. The authors should conduct a thorough review of existing research, highlighting relevant studies and situating their work within the broader academic context.

Experimental design

4- The methodology section would benefit from the inclusion of a flow diagram. This should illustrate the overall research process and clearly indicate where the authors' contributions fit within this framework.

Validity of the findings

5- The results section is missing details about the evaluation metrics used. The authors should clearly describe the metrics employed to assess their work, including any rationale for their selection.

6- There is a lack of information about the datasets used in the study. The authors should include comprehensive details about the datasets, including their sources, characteristics, and any preprocessing steps undertaken.

Additional comments

7- The paper does not currently have a discussion section. The authors should add this to interpret their results, discuss their implications, and suggest potential areas for future research.

8- The objectives of the research are not clearly stated. The authors should provide a concise summary of their research goals early in the paper to guide readers through their study.

9- Both the abstract and conclusion sections are lacking in detail. The authors should ensure that the abstract succinctly summarizes the key points of their research, and that the conclusion effectively synthesizes their findings and contributions.

10- The paper should be reviewed for consistency and coherence. This includes ensuring that terminology is used consistently throughout, that arguments are logically structured, and that the overall narrative flows smoothly from one section to the next.

Cite this review as

Reviewer 2 ·

Basic reporting

Dear Author,
Thank you for submitting your manuscript titled "Automated Programming Approaches to Enhance Computer-Aided Translation Accuracy" . I appreciate your detailed exploration of the advancements in computer-aided translation (CAT), particularly focusing on the integration of automatic programming to improve translation accuracy. Your work is timely and relevant given the growing reliance on CAT systems in the globalized world. Based on the initial evaluation, I recommend that your manuscript undergoes major revisions to enhance its clarity, depth, and methodological rigor. Please find detailed comments below that should guide your revisions.
Recommendations: Major Revision
1. While the purpose of the study is clearly stated in the abstract, the specific research questions or hypotheses are not clearly articulated. Please clarify these elements to better frame the study's goals.

Experimental design

2. The methodology section would benefit from a more detailed description of the CAT systems used, particularly the specific algorithms and translation memory technologies employed (e.g., pages 3–4). This will aid in assessing the study's replicability.
3. On page 6, the metrics used to evaluate translation accuracy are introduced, but their selection criteria and relevance to the study's objectives are not discussed in depth. Please provide a rationale for choosing these metrics.
4. The manuscript would benefit from a more robust statistical data analysis, particularly when comparing CAT and traditional methods. Consider including tests of significance and confidence intervals to support your findings (page 7).
5. The concept and application of translation memory technology are mentioned briefly. It would be useful to expand on how these memories were built and utilized, as well as their impact on translation accuracy.
6. In the section on automatic programming, more detail is needed on how these programs specifically enhance the translation process compared to manual programming.
7. Figures 1 and 2, which depict the CAT model and translation memory, should be more clearly labeled and referenced in the text. Additionally, the descriptions should elaborate on their relevance to the study's findings.
8. Provide more information about the corpus used for testing CAT systems, including details on size, content, and selection criteria. This is crucial for understanding the study's scope and limitations.

Validity of the findings

9. Address how the complexity of the source text (e.g., technical jargon, acronyms) might affect translation accuracy differently for CAT and human translators.
10. Discuss how CAT systems handle contextual nuances in translation compared to human translators. Are there specific strategies or tools used?
11. Strengthen the conclusion by summarizing the key findings and their implications for the future of CAT and translation studies.

Additional comments

none

Cite this review as

---

## Round 0.2 · Minor Revisions

Dear authors,

Thank you for re-submitting your manuscript with the incorporation of technical comments. I am pleased to inform you that reviewers are now satisfied with the technical contribution of your manuscript. To uphold the further quality of the paper. I suggest you add the future research direction and improve the langue of the paper in the second round as it would be a good time to improve the paper's language.

Thank you

Reviewer 1 ·

Basic reporting

All changes have been completed.

Experimental design

All changes have been completed.

Validity of the findings

All changes have been completed.

Cite this review as

Reviewer 2 ·

Basic reporting

Thank you for addressing the comments. I would recommend adding 'Future Research Directions' and a double read for English improvements.

Experimental design

N/A

Validity of the findings

N/A

Additional comments

N/A

Cite this review as

---

## Round 0.3 · accepted · Accept

Thanks for your resubmission, after careful consideration of the revised version, I'm pleased to notify you that we are now recommending your article for publication. Thank you for your fine contribution.